# Charge Accumulation of Amplified Spontaneous Emission in a Conjugated Polymer Chain and Its Dynamical Phonon Spectra

**DOI:** 10.3390/molecules25133003

**Published:** 2020-06-30

**Authors:** Zhe Lin, Jiahao Chen, Yusong Zhang, Jianguo Shen, Sheng Li, Thomas F. George

**Affiliations:** 1State Key Laboratory of Surface Physics, Department of Physics, Fudan University, Shanghai 200433, China; 17210190005@fudan.edu.cn (Z.L.); 16110190015@fudan.edu.cn (Y.Z.); 2Department of Physics, Zhejiang Normal University, Jinhua 321004, China; chenjh1218@gmail.com (J.C.); shenjianguo@zjnu.cn (J.S.); shenglee@zjnu.cn (S.L.); 3Department of Chemistry &Biochemistry and Physics & Astronomy, University of Missouri–St. Louis, St. Louis, MO 63121, USA

**Keywords:** amplified spontaneous emission, spontaneous electric polarization, conjugated polymers

## Abstract

In this article, the detailed photoexcitation dynamics which combines nonadiabatic molecular dynamics with electronic transitions shows the occurrence of amplified spontaneous emission (ASE) in conjugated polymers, accompanied by spontaneous electric polarization. The elaborate molecular dynamic process of ultrafast photoexcitation can be described as follows: Continuous external optical pumping (laser of 70 µJ/cm^2^) not only triggers the appearance of an instantaneous four-level electronic structure but causes population inversion for ASE as well. At the same time, the phonon spectrum of the conjugated polymer changes, and five local infrared lattice vibrational modes form at the two ends, which break the original symmetry in the system and leads to charge accumulation at the ends of the polymer chain without an external electric field. This novel phenomenon gives a brand-new avenue to explain how the lattice vibrations play a role in the evolution of the stimulated emission, which leads to an ultrafast effect in solid conjugated polymers.

## 1. Introduction

The first success of producing a lasing effect in a conjugated polymer solution was achieved in 1992 [1]. Then, with the progress in solid polymeric light-emitting diodes (PLEDs) [2,3], solid conjugated polymeric lasers were realized in sundry materials [4,5,6,7]. Due to the importance of the resonator in lasing, many experiments focus on extending the list of candidates for resonators: distributed Bragg reflectors [8,9,10,11,12], 2D photonic crystals [13,14,15,16], photonic bandgap fibers [17], whispering galleries [18,19,20], feedback structures [21,22,23], or even flexible polymer fibers [24].

On the other side, theoretical researchers have been paying considerable attention to the mechanism behind that. However, there are still some ambiguities that need to be clarified. For example, considering the fact that the decay of excitons (electron–hole pairs) leads to the luminescence of a PLED, some researchers sometimes think that the mechanism of polymeric lasing/amplified spontaneous emission (ASE) is similar to the luminescence of a PLED. However, lasing/ASE has to satisfy a strict condition—electron population inversion—which does not exist in a PLED. The mechanics of optically pumped conjugated polymeric lasing/ASE can be briefly depicted as follows: Given a three- or four-level energy structure, continuous external optical pumping leads not only to excitons but also to electron population inversion. Some initial spontaneously emitted photons caused by excitons greatly stimulate other light emission because of population inversion, and then brings about lasing, or ASE, if without a resonator. We will fully explain the mechanics in this article.

Since Y. R. Shen et al. uncovered the self-trapping effect in conjugated poly(p-phenylene vinylene) (PPV) as originating from the vibrational-electronic double resonance [25], the phonons in conjugated polymers have been attracting attention, since intramolecular self-trapping opens up an avenue to spatially track the excited-state dynamics along the polymer chain by phonons [26]. In 2009, the optically pumped organic semiconductor laser was found to be largely dependent on the microscopic ultrafast dynamics [27]. In 2014, an exciting work showed at the beginning of the external optical pumping that the stimulated emission of the conjugated polymer is easily influenced by a multiple phonon process [28]. Besides, it was reported that ultrafast energy relaxation in low-dimensional π-electrons is also dominated by phonon emission once excitation occurs [29]. In spite of recent research which has explored the dynamical process of population inversion of a conjugated polymer [30], research seldom involves the dynamical phonon spectra and electron transitions. It thus becomes necessary to develop a valid dynamics method to investigate the ultrafast process of ASE and discover the relationship between phonon spectra and electron transitions, which is a key purpose of this our article.

During the non-adiabatic dynamics of a conjugated polymer, it has been found that electronic transitions connect to the localization of the electron, while the high-frequency phonon vibrational modes are closely related to the evolution of the molecular orbitals [31]. Thus, a natural assumption or anticipation is proposed here: Is ASE in a conjugated polymer, with the aid of lattice vibrational modes, able to redistribute the charge along the polymer chain?

To clarify this, the dynamical process of ASE, involving the dynamical phonon spectrum and charge distribution, must be investigated. Fortunately, with a model Hamiltonian, we are able to present a dynamical approach that not only incorporates the electron transition process with non-adiabatic dynamics but also links the dynamical phonon spectra with ultrafast dynamics. All of this makes it possible to understand the dynamical evolution with regard to stimulated emission, electronic structure, lattice structure, and charge distribution of conjugated polymers. On the basis of this, it is promising to delineate how the phonon vibrational modes influence the evolution of the stimulated emission and what the new behavior is for ASE.

Recently, our research with respect to ASE of a conjugated polymer [32] has explored the static phonon spectrum and electronic structure, showing all vibration modes are symmetric. Interestingly, as we try to get closer to the real situation and then greatly increase the non-radiative part of the electron transition process, vibrational modes turn out to be asymmetric, and charge will accumulate at the two ends of the chain.

## 2. Results and Discussion

When a single PPV chain with 200 unit clusters is stimulated by an external optical field (70 µJ/cm^2^) with proper frequency, the electrons will jump among different levels, namely electronic transitions start.

Figure 1 shows the evolution of the energy of the relevant levels, where we use A, B, C and D to label sub-LUMO, LUMO, HOMO, and sub-HOMO. Once electrons in level D are light-pumped to A, these four levels become distinguishable as they quickly slip into the gap between B (HOMO) and C (LUMO) in the first 100 fs, then stabilizing at 400 fs. These four discrete energy levels eventually meet the prerequisite conditions for ASE or lasing. Especially, the energies of Levels A and B, or Levels C and D, are quite close, indicating very strong non-radiative behavior. As a result, the pumped electrons in Level A can easily jump down to Level B, and the electrons can also easily slip to Level D, which makes this four-level system amenable to population inversion.

Naturally, we now turn our attention to the evolution of the electron population of the involved levels as delineated in Figure 2. In Figure 2a, let us assume *n_2_* as the population of level B and *n_1_* as that of level C. Under the pumping by an external optical field, the electron (spin-up) population of the original LUMO, namely level B, grows from 0.00 to 0.63 within 600 fs. On the other hand, the electron (spin-up) population of the original HOMO (level C) diminishes from 1.00 to 0.42. We notice the population crossing point of Levels B and C is 102 fs, which indicates a population inversion between them. The Final stable population distributions is shown in Figure 2b, which clearly illustrates the population inversion between level B and C in the end.

Between levels B and C, the spontaneous emission initially lets a few electrons in level B drop to the lower energy level C and then emits light. Apparently, once *n_2_* > *n_1_* occurs, the population inversion makes the emitted optical intensity much stronger. This process is the so-called amplified spontaneous emission (ASE). Thus, the occurrence of electron population inversion means the appearance of ASE. Integrated with the final electronic energy structure and the electron (spin-up) population, Figure 2b illustrates that once the time exceeds 102 fs, the electron population of level B becomes larger than C, indicating the occurrence of ASE in the conjugated polymer.

Considering the strong coupling between the electrons and lattice in a conjugated polymer, such as PPV, it also has been reported that the formed four-level structure that contributes to ASE/lasing is tightly associated with lattice vibrations [33]. Thus, we undertake a further investigation of the vibrational modes during the process of photoexcitation.

After the studied conjugated polymers undergo pumping by an external laser with intensity of 70 µJ/cm^2^ from the beginning until 1 ps, the phonon spectrum changes accordingly. Figure 3 represents the dynamical phonon spectra at the seven different times of 0 fs, 10 fs, 20 fs, 30 fs, 40 fs, 50 fs and 1 ps in which the component of vibrational mode can be obtained by projecting the time-dependent lattice configuration onto the eigenvectors (vibrational modes/phonon modes). Although it seems to be only one peak at the beginning of the photoexcitation, there are actually two peaks very close to each other. After that, within 50 fs, several more peaks begin to appear. Comparing the phonon spectra at the final time (1 ps) with the beginning (0 fs), we find that the highest peak of 939.2 cm^−1^ in the spectra decreases dramatically.

The appearance of the new peaks in the phonon spectrum after the photoexcitation indicates the change of lattice oscillations. The lattice vibration modes at the time of 1 ps are marked by M1 through M7 in Figure 4. Due to the excitation by an external optical field of 70 μJ/cm^2^, the sites of the local lattice vibrational modes move to the two ends of the polymer chain, indicating that the excitation of 70 μJ/cm^2^ finally produces symmetry breaking.

More results can be obtained if we compare this symmetry breaking with our previous research [32], where vibrational modes are all symmetric. This symmetry breaking also reflects the significant role played by the non-radiative transition. Once the rate of the non-radiative transition is tuned to a real situation, e.g., 10^13^/s, the non-radiative transition will overcome the original localization to break away from the original symmetry.

Up to 1 ps, we observe several more scattered peaks in Figure 5 as compared with the beginning of the optical pumping. At 1 ps, both the highest and second highest peaks in the photon spectrum reflect the background of the alternating bonds of PPV, which are contributed to by the lattice vibrational modes M1 and M2 as in Figure 4a; these modes are indeed extensive. However, there are still five lattice vibrational modes in Figure 4b–f, along with the continuous external optical pumping, and all of them possess even parity. This means that these five modes are not only localized but are infrared lattice vibrational modes as well.

While the lattice vibrational modes M1 and M2 contribute to peaks 1 and 2 in the phonon spectrum, the new lattice vibrational modes, such as in Figure 4b–f, contribute to the new resultant peaks in the phonon spectrum. We see that these five localized lattice vibrational modes are not localized at the center but move to the ends of the polymer chain, which leads to the breaking of the original symmetry of the related physical properties. We thus direct our attention to the evolution of the charge density within 1 ps (Figure 6 and Figure 7). Correspondingly, along with the evolution of the dynamical phonon spectra in Figure 3, Figure 6 illustrates the charge distribution at the six different times of 0 fs, 10 fs, 20 fs, 30 fs, 40 fs, and 1 ps. As the electronic energy structure and electron population undergo significant oscillations in the first 50 fs, from Figure 6d,e, the charge distribution begins to oscillate from 30 to 40 fs. From 30 to 40 fs, serious lattice oscillations significantly change the phonon spectrum as shown in Figure 2 (30 fs, 40 fs), and then the changing phonon spectrum, coupled with electron–phonon interaction, strongly drives the oscillation of charge distribution, as shown in Figure 6d,e.

After ASE appears under photoexcitation by an optical field of intensity 70 μJ/cm^2^, the fluctuation continues but slows down from the time-dependent charge distribution within the first 1 ps, as seen in Figure 7. At 300 fs, the occurrence of the local lattice vibrational modes results in symmetry breaking. The positive charge then is localized over 30–70 units along the PPV chain, while the negative one is over 130–170 units. The net charge starts to remain steady, which also leads to the new charge distribution along the chain being destroyed during the photoexcitation. During the occurrence of ASE in a conjugated polymer, a significant phenomenon of charge accumulation at the ends of the PPV chain occurs without an external electric field. Two accumulation positive/negative regions here coincide with the localized regions of the lattice vibrational modes as depicted in Figure 4b–f.

## 3. Methods

### 3.1. Hamiltonian

Based on earlier semiconducting research on carrier in photoexcited conjugated polymers [30,32,33,34], poly(p-phenylene vinylene) (PPV) is selected as a representative to investigate dynamics of polymeric molecule. For the Hamiltonian, not only has the quasi-one-dimensional structure been taken into account, but also the strong interactions have been considered in the system:(1)$$H=H_{e}+H_{l}+H{}^{\prime }.$$
Here, *H_e_ + H_l_* is the well-known Su–Schrieffer–Heeger Hamiltonian [35], which is used to describe the physical properties of conjugated polymers. The first part, *H_e_*, describes the electron–lattice interaction,
(2)$$H_{e}=-\sum_{l,s}\left[t_{0}-\alpha \left(u_{l+1}-u_{l}\right)+{\left(-1\right)}^{l}t_{e}\right]\left(a_{l+1,s}^{+}a_{l,s}+H.c.\right),$$
where the parameters are explained as follows: *t_0_* (2.5–3.0 eV) is a hopping constant; *t_e_* (0.05–0.10 eV) is the Brazovskii–Kirova term [36]; *α* (4.0–6.0 eV/Å) is an electron–lattice interaction constant; and $a_{l,s}^{+}/a_{l,s}$ is the electron creation/annihilation operator.

*H_l_* is for the kinetic energy and the elastic potential energy of the polymer lattice,
(3)$$H_{l}=\frac{M}{2}\sum_{l}{\left({\dot{u}}_{l}\right)}^{2}+\frac{K}{2}{\sum_{l}\left(u_{l+1}-u_{l}\right)}^{2},$$
where *K* (20–30 eV/Å^2^) is an elastic constant, and *M* is the group mass. *H’* represents the electron–electron interaction, which is actually the extended Hubbard electron interaction [37]:(4)$$H{}^{\prime }=U\sum_{l}n_{l,↑}n_{l,↓}+V\sum_{l,s,s{}^{\prime }}n_{l,s}n_{l+1,s{}^{\prime }}.$$
The parameter *U* (2.0–5.0 eV) is the on-site Coulomb interaction, while *V* (0.5–2.0 eV) is the nearest-neighbor Coulomb interaction.

For convenience of computation, the electron–electron interaction *H’* is treated within the Hartree–Fock approximation as
(5)$$\begin{array}{l} H{}^{\prime }=\sum_{l,s}\left\{U\left(\sum_{\mu }^{occ}{\left|Z_{l,\mu }^{-s}\right|}^{2}-\frac{1}{2}\right)+V\left[\sum_{s{}^{\prime }}\left(\sum_{\mu }^{occ}{\left|Z_{l+1,\mu }^{-s{}^{\prime }}\right|}^{2}+\sum_{\mu }^{occ}{\left|Z_{l+1,\mu }^{-s{}^{\prime }}\right|}^{2}-2\right)\right]\right\}a_{l,s}^{+}a_{l,s}\\ -\sum_{l,s}\left(V\sum_{\mu }^{occ}Z_{l,\mu }^{s}Z_{l+1,\mu }^{s}\right)a_{l,s}^{+}a_{l,s}+H.c. \end{array}$$
where *occ* is the occupation number.

The wavefunction with spin *s* can be written as
(6)$$\begin{array}{l} \varepsilon_{\mu }Z_{l,\mu }^{s}=\left[U\left(\rho_{l}^{-s}-\frac{1}{2}\right)+V\left(\sum_{s{}^{\prime }}\rho_{l-1}^{s{}^{\prime }}+\sum_{s{}^{\prime }}\rho_{l+1}^{s{}^{\prime }}-2\right)+Ee\left(l-\frac{N+1}{2}\right)a\right]Z_{l,\mu }^{s}\\ -\left[V\sum_{\mu }^{occ}Z_{l,\mu }^{s}Z_{l-1,\mu }^{s}+t_{0}+\alpha \left(u_{l-1}-u_{l}\right)+{\left(-1\right)}^{l-1}t_{e}\right]Z_{l-1,\mu }^{s}\\ -\left[V\sum_{\mu }^{occ}Z_{l,\mu }^{s}Z_{l+1,\mu }^{s}+t_{0}+\alpha \left(u_{l+1}-u_{l}\right)+{\left(-1\right)}^{l+1}t_{e}\right]Z_{l+1,\mu }^{s} \end{array}$$
and the distribution of charge can be obtained by $\rho_{l}=\sum_{\mu }^{occ}{\left|\Psi_{l,\mu }\right|}^{2}-n_{0}$, where *n_0_* is the density of the positively-charged background.

### 3.2. Non-Adiabatic Dynamics

For non-adiabatic dynamics, the time-dependent Schrödinger equation is employed directly:(7)$$i\hbar \frac{\partial }{\partial t}\Psi =H\Psi .$$

If the initial electronic wavefunction is $\Psi \left(t_{i}\right)$ and the final one is $\Psi \left(t_{f}\right)$, the time-independent Hamiltonian during a very short span of time ($t_{f}-t_{i}$) is
(8)$$\Psi \left(t_{f}\right)=\exp\left\{-\frac{iH}{\hbar }\left(t_{f}-t_{i}\right)\right\}\Psi \left(t_{i}\right).$$

Beginning with *t_1_*, *t_2_* (until *t_N_*), the electronic wavefunction of the later time is the time evolutional result of the wavefunction at the earlier time. If the interval is quite short, like 0.01 fs, the wavefunction at any time can be acquired. Then the electronic wavefunction at any time can be calculated by a linear combination:(9)$$\Psi \left(t\right)=\sum_{\mu }c_{\mu }\left(t\right)\Phi_{\mu }.$$
Here $\Phi_{\mu }$ is an eigenfunction. Due to the orthonormality of the eigenfunctions, the expansion coefficients can be written as
(10)$$c_{\mu }\left(t\right)=\langle \Phi_{\mu }|\Psi \left(t\right)\rangle .$$

The spontaneous emission between two levels (A and B as example) occurs at a rate:(11)$$\gamma_{AB}=\frac{4{\left(E_{A}-E_{B}\right)}^{3}}{3h^{4}c^{3}}p^{2},$$
while the rate of stimulated emission is
(12)$$\lambda_{AB}=\frac{\pi }{3\varepsilon_{0}h^{2}}p^{2}\rho \left(\omega_{0}\right).$$
Here, p=e〈A|r|B〉 denotes the transition dipole moment between the two energy levels, and $\rho \left(\omega_{0}\right)$ is the energy density of the external photoelectric field.

Together with Equations (11) and (12), if non-radiative transition is at a rate *k*~10^13^/s, the electron transition process among four levels (A–D) can be depicted as following:(13)$$\left\{ \begin{array}{l} \frac{dP_{A}}{dt}=W_{DA}P_{D}-W_{AD}P_{A}-\gamma_{AB}P_{A}-k_{AB}P_{A}\\ \frac{dP_{B}}{dt}=\gamma_{AB}P_{A}-\gamma_{BC}P_{B}+k_{AB}P_{A}-k_{BC}P_{B}\\ \frac{dP_{C}}{dt}=\gamma_{BC}P_{B}-\gamma_{CD}P_{C}+k_{BC}P_{B}-k_{CD}P_{C}\\ \frac{dP_{D}}{dt}=n-P_{A}-P_{B}-P_{C} \end{array}\right.$$
With the help of the Feynman–Hellmann theorem,
(14)$$F_{l}=-\langle \Psi |\frac{\partial H}{\partial u_{l}}|\Psi \rangle ,$$
we ultimately arrive at an equation describing the motion of lattice:(15)$$M\frac{d^{2}u_{l}}{dt^{2}}=-\sum_{\mu }^{occ}\frac{\partial \varepsilon_{\mu }}{\partial u_{l}}+K\left(2u_{l}-u_{l+1}-u_{l-1}\right).$$

After combining the transition process with the non-adiabatic electronic transition equations, it becomes possible to describe the ultrafast dynamics of amplified spontaneous emission in conjugated polymers.

### 3.3. Lattice Vibrational Modes and Phonon Spectrum

Lattice vibrations contribute to the localization in conjugated polymers. To seek approximations for vibrational modes, the perturbation method should be introduced. Firstly, if the static lattice configuration is $\phi_{n}^{0}$ and tiny vibration around $\phi_{n}^{0}$ is $\phi {{}^{\prime }}_{n}\left(t\right)$, the real configuration is:(16)$$\phi_{n}\left(t\right)=\phi_{n}^{0}+\phi {{}^{\prime }}_{n}\left(t\right).$$

We then expand the expression of energy with a second-order perturbation of $\phi {{}^{\prime }}_{n}\left(t\right)$:(17)$$H\left(\left\{\phi_{n}\right\}\right)=E_{0}+E_{s}\sum_{m}^{N}A_{m}\left(\left\{\phi_{n}\right\}\right)\phi {{}^{\prime }}_{m}+\frac{1}{2}\sum_{m,n}^{N}B_{m,n}\left(\left\{\phi_{n}^{0}\right\}\right)\phi {{}^{\prime }}_{m}\phi {{}^{\prime }}_{n}$$
(18)$$\begin{array}{l} B_{m,n}=k[\left(\delta_{m,n}+\delta_{m,n+1}\right)(1-\delta_{m,N})+\left(\delta_{m,n}+\delta_{m,n-1}\right)(1-\delta_{n,1})]\\ +2\alpha^{2}{\left(-1\right)}^{m+n}\sum_{\mu ,\nu \left(\neq \mu \right)}\frac{C_{\mu ,\nu }^{m}C_{\mu ,\nu }^{n}}{\varepsilon_{\mu }^{0}-\varepsilon_{\nu }^{0}}. \end{array}$$

N is the total number of lattice sites, *ε^0^* is the static eigenenergy of the electron, and the coefficient $C_{\mu ,\upsilon }^{m}$ is
(19)$$\begin{array}{l} C_{\mu ,\nu }^{m}=\left(1-\delta_{m,N}\right)\left(Z_{\mu ,m+1,s}^{0}Z_{\nu ,m,s}^{0}+Z_{\mu ,m,s}^{0}Z_{\nu ,m+1,s}^{0}\right)\\ -\left(1-\delta_{m,1}\right)\left(Z_{\mu ,m,s}^{0}Z_{\nu ,m-1,s}^{0}+Z_{\mu ,m-1,s}^{0}Z_{\nu ,m,s}^{0}\right). \end{array}$$
Here, $Z_{\mu ,m,s}^{0}$ is the corresponding static eigenstate of the electron, and $\delta_{m,n}$ is the Kronecker delta. As long as we diagonalize matrix **B**, its eigenvalue can be used to calculate the frequency, and the eigenvector shows how the lattice vibrates. Consequently, the excitation of lattice vibration modes during the formation of ASE can be depicted in detail.

## 4. Conclusions

A detailed process of charge accumulation of ASE in conjugated polymers after photoexcitation is uncovered. An external laser (70 µJ/cm^2^) induces the occurrence of an instantaneous four-level electronic structure within 400 fs and population inversion, both of which are responsible for ASE. The phonon spectrum in the system also changes at the same time. It is seen that the resultant five new localized lattice vibrational modes break the original symmetry in the first 100 fs. This symmetry breaking, without the help of an external electric field, is closely related to the non-radiative transition and ultimately leads to charge accumulation at the ends of the polymer chain. Even parity in these five localized lattice vibrational modes means that they can be probed by the infrared phonon spectrum. Overall, this article opens the opportunity to realize this ultrafast process as associated with solid conjugated polymers.

## Figures and Tables

**Figure 1 molecules-25-03003-f001:**
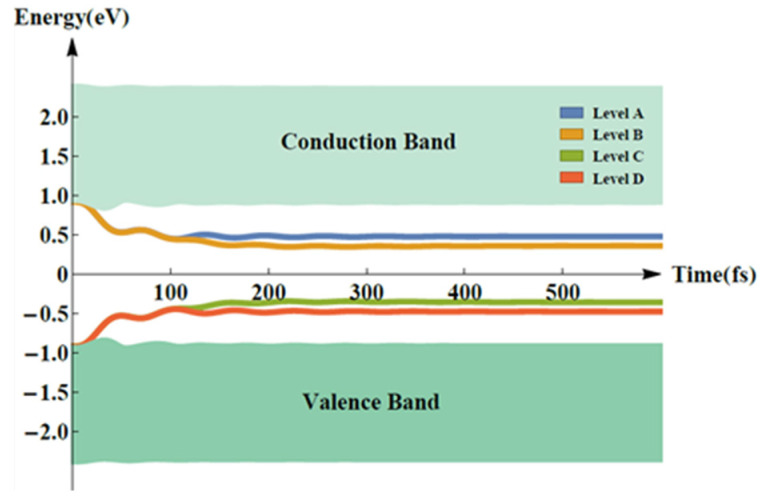
Evolution of the energy levels within 500 fs under pumping by an external laser of 70 µJ/cm^2^.

**Figure 2 molecules-25-03003-f002:**
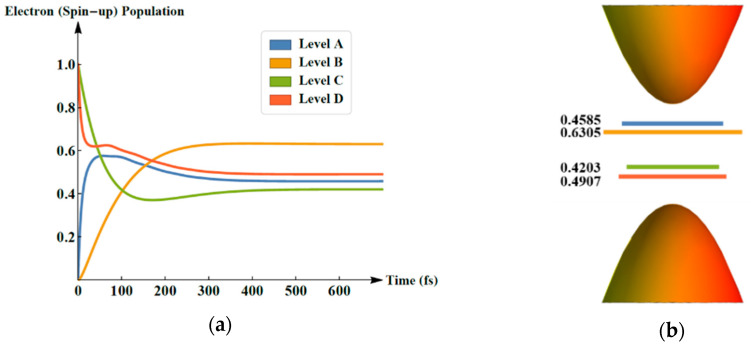
(**a**) Evolution of the electron (spin-up) population within 600 fs; (**b**) final stable population distributions.

**Figure 3 molecules-25-03003-f003:**
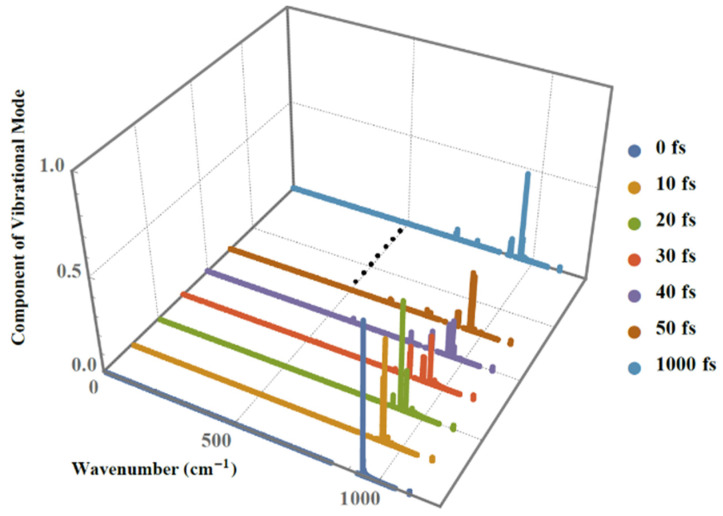
Dynamical phonon spectrum during the formation of amplified spontaneous emission (ASE) under pumping by an external laser of 70 µJ/cm^2^.

**Figure 4 molecules-25-03003-f004:**
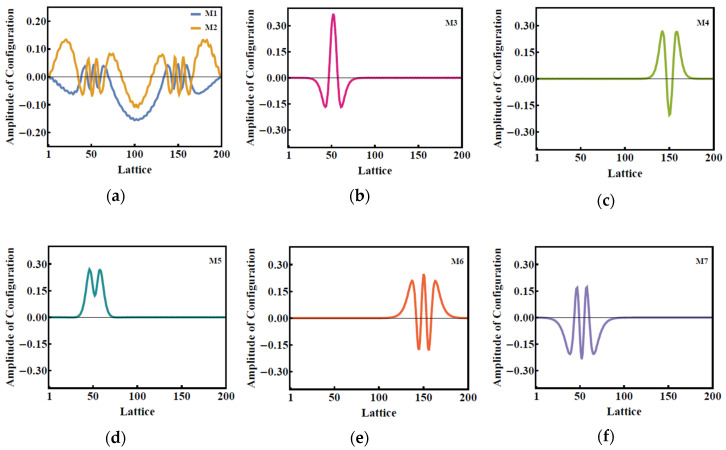
(**a**) Extensive lattice vibrational modes and (**b**–**f**) local lattice vibrational modes under pumping by an external laser of 70 µJ/cm^2^.

**Figure 5 molecules-25-03003-f005:**
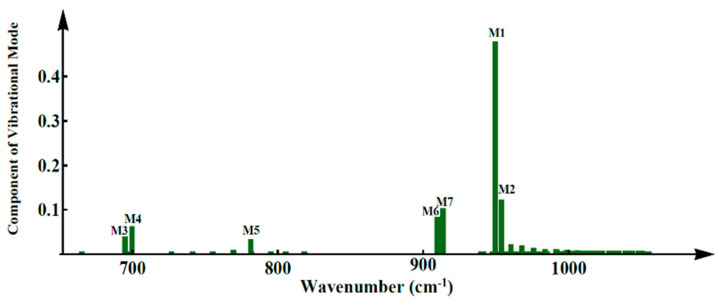
Phonon spectra of ASE under pumping by an external laser of 70 µJ/cm^2^.

**Figure 6 molecules-25-03003-f006:**
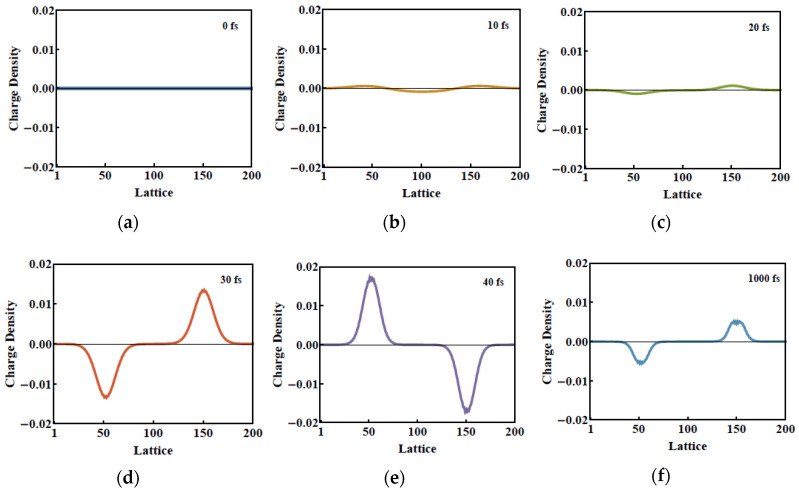
(**a**–**f**) Evolution of the charge density distribution along the poly(p-phenylene vinylene) (PPV) polymer chain by an external laser of 70 µJ/cm^2^ at 0 fs, 10 fs, 20 fs, 30 fs, 40 fs, and 1 ps.

**Figure 7 molecules-25-03003-f007:**
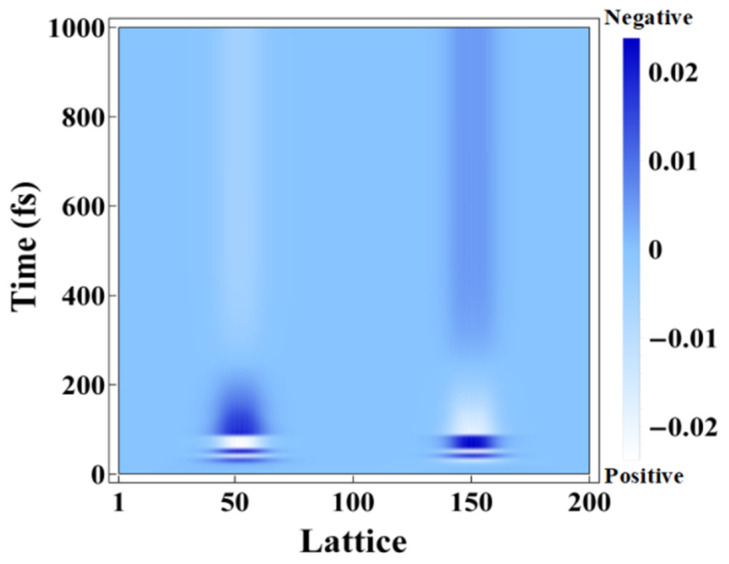
Evolution of the charge density distribution along the PPV polymer chain by an external laser of 70 µJ/cm^2^.

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
