# Peer review of "Charge Accumulation of Amplified Spontaneous Emission in a Conjugated Polymer Chain and Its Dynamical Phonon Spectra"

_molecules, 2020, doi:10.3390/molecules25133003_

Round 1
Reviewer 1 Report
The work is written clearly. The results are excellent.
Author Response
We appreciate the recommendation by this reviewer that our manuscript be published in Molecules without revisions. Accordingly, we have made no changes based on this report.
The authors
Reviewer 2 Report
The authors in this paper describe the phonon spectrum in a well known conjugated polymer (PPV) after pump excitation. They describes this process durig the ASE effect.
Unfortunately I do not agree the initial approach made by the authors , they speak about population inversion between the HOMO and LUMO after excitation, and they call this the main step for Amplified Spontaneous Emission. In my point of view this is the first step for an exciton creation and then spontaneous emission. The ASE effect happens when we excite a large area of the sample and the first spontaneous emitted photon stimulates the emission ofother ones, so to bring to an amplification.
They write:" For example, influenced by the fact that excitons (electron-hole pairs) lead to the luminescence of a PLED, many
researchers also attribute polymeric lasing/ASE to excitonic behavior. Unfortunately, lasing/ASE needs electron population inversion while excitons fail to explain it." The exciton creation is already as a matter of fact a population inversion: the electron from the HOMO level is promoted to the LUMO one. This is the inital step to excite a molecule, to create an exciton. This mechanism can also brings to free charges formation, but the free charges do not bring to emission.
Based on this argument I can not suggest the publication of this manuscript.
Author Response
Thank you very much for the most helpful report of Reviewer 2. We have revised the manuscript accordingly. We feel that it is much improved by our revisions, which we would like to discuss below. Our major changes are highlighted in yellow both here and in the revised manuscript.
Reviewer 2’s Comments and Authors’ Responses/Revisions
Comment: Unfortunately, I do not agree the initial approach made by the authors, they speak about population inversion between the HOMO and LUMO after excitation, and they call this the main step for Amplified Spontaneous Emission. In my point of view this is the first step for an exciton creation and then spontaneous emission. The ASE effect happens when we excite a large area of the sample and the first spontaneous emitted photon stimulates the emission of other ones, so to bring to an amplification.
They write:" For example, influenced by the fact that excitons (electron-hole pairs) lead to the luminescence of a PLED, many researchers also attribute polymeric lasing/ASE to excitonic behavior. Unfortunately, lasing/ASE needs electron population inversion while excitons fail to explain it." The exciton creation is already as a matter of fact a population inversion: the electron from the HOMO level is promoted to the LUMO one. This is the initial step to excite a molecule, to create an exciton.
Response: Here, it has to been emphasized that the Amplified Spontaneous Emission is totally different from the Spontaneous Emission, even though the two terminologies have the same words in term of Spontaneous Emission. Amplified Spontaneous Emission is the behavior of stimulated emission which has been presented in various textbooks, especially regarding lasing mechanism, while Spontaneous Emission is attributed only to the decay of the conventional exited state ‒ exciton. What’s more, the exciton can be created once the electron from the HOMO level is promoted to the LUMO without being population inversion.
In the revised version of the manuscript, in order to eliminate this misconception, we have provided some related sentences in the Introduction, which are highlighted in yellow as follows:
For example, considering the fact that the decay of excitons (electron-hole pairs) leads to the luminescence of a PLED, some researchers sometimes think that the mechanism of polymeric lasing/ASE is similar to the luminescence of a PLED. However, lasing/ASE has to satisfy a strict condition ‒ electron population inversion ‒ which doesn't exist in a PLED. The mechanics of optically-pumped conjugated polymeric lasing/ASE can be briefly depicted as follows: Given a three- or four-level energy structure, continuous external optical pumping leads not only to excitons but also to electron population inversion. Some initial spontaneous emitted photons caused by excitons greatly stimulates other light emission because of population inversion, and then brings about lasing, or ASE if without a resonator. We will fully explain the mechanics in this article.
Comment: This mechanism can also bring to free charges formation, but the free charges do not bring to emission.
Response: In polymeric solar cells, the decomposition of an exciton (electron-hole pair) can lead to the formation of free charges. In the article, the charged accumulation in the ASE is not so-called free charges, but rather the localized and excited state are characterized by the symmetry breaking of the stimulated emission.Sincerely,
The authors
Reviewer 3 Report
In the manuscript “Charge Accumulation of Amplified Spontaneous Emission in a Conjugated Polymer Chain and Its Dynamical Phonon Spectra” the authors continue their theoretical studies concerning amplified spontaneous emission. The work will be acceptable if the following points will be better clarified.
- The four level model for PVV had been already described in reference 32 along with localized symmetrical phonon vibrational modes; in the present case, however, localized phonons are obtained which unbalance charge distribution during excitation. Does this effect depend on the difference in optical pumping between the present work and the first one? depending on an optical pumping with higher energy than the energy of HOMO LUMO gap? In the manuscript, comparison is not explicitly conducted between the two situations and the authors do not suggest reasons for the differences from previous results. Is there an energy threshold for symmetry breaking?
- A discussion/explanation should be given regarding the physical reason for the charge distribution oscillations occurring from 30 to 40 fs.
- The theoretical model has been used in many instances in the literature, however precise citations should be given referring to authors that effectively contributed to the model: Su-Schrieffer-Heeger Hamiltonian: cite the original work or a review of the authors; also regarding Brazovskii-Kirova term, Hubbard electron interaction, …. please refer to the appropriate literature. (by the way it is Schrieffer, not Schreiffer).
Minor corrections, commenting figure 2 “We notice the crossing point of levels B and C in is 102 fs…” should be “ We notice the population crossing point of levels B and C is 102 fs…”
Please, illustrate the role of levels A and D in obtaining population inversion between levels B and C
Page 4 line 111: “After conjugated polymers undergo pumping by an external laser with intensity of 70 μJ/cm2..” should be “After the studied conjugated polymers undergo pumping by an external laser with intensity of 70 μJ/cm2….”
Better specify the meaning of: “component of vibrational mode” in figure 3 and 5.
Author Response
Reviewer 3’s Comments and Authors’ Responses/Revisions
Comment: The four-level model for PVV had been already described in Reference 32 along with localized symmetrical phonon vibrational modes; in the present case, however, localized phonons are obtained which unbalance charge distribution during excitation. Does this effect depend on the difference in optical pumping between the present work and the first one? depending on an optical pumping with higher energy than the energy of HOMO LUMO gap? In the manuscript, comparison is not explicitly conducted between the two situations and the authors do not suggest reasons for the differences from previous results. Is there an energy threshold for symmetry breaking?
Response: For the differences between this present article and Reference 32, both intensity of light and non-radiative transition play key roles. In Reference 32, the the intensity of light is 60 μJ/cm2. In this article, the non-radiative transition increases to 1013/s when the intensity of pumping light is tuned to 70 μJ/cm2. Therefore, the non-radiative transition in this process is strong enough that the system breaks away from the original symmetry. More comparison and discussion are added in the revised manuscript as follows:
In the Introduction:
To clarify this, the dynamical process of ASE, involving the dynamical phonon spectrum and charge distribution, must be investigated. Fortunately, with a model Hamiltonian, we are able to present a dynamical approach that not only incorporates the electron transition process with non-adiabatic dynamics, but also links the dynamical phonon spectra with ultrafast dynamics. All of this makes it possible to understand the dynamical evolution with regard to stimulated emission, electronic structure, lattice structure and charge distribution of conjugated polymers. On the basis of this, it is promising to delineate how the phonon vibrational modes influence the evolution of the stimulated emission and what the new behavior is for ASE.
Recently, our research with respect to ASE of a conjugated polymer [32] has explored the static phonon spectrum and electronic structure, showing all vibration modes are symmetric. Interestingly, as we try to get closer to the real situation and then greatly increase the non-radiative part of the electron transition process, vibrational modes turn out to be asymmetric, and charge will accumulate at the two ends of the chain.
Below Figure 4 add:
More results can be obtained if we compare this symmetry breaking with our previous research [32], where vibrational modes are all symmetric. This symmetry breaking also reflects the significant role played by the non-radiative transition. Once the rate of the non-radiative transition is tuned to a real situation, say 1013/s, the non-radiative transition will overcome the original localization to break away from the original symmetry.
Comment: A discussion/explanation should be given regarding the physical reason for the charge distribution oscillations occurring from 30 to 40 fs.
Response: We thank the reviewer for this suggestion, and more explanation has been inserted above Figure 6 as follows:
Above Figure 6 add:
From 30 to 40 fs, serious lattice oscillations significantly change the phonon spectrum as shown in Fig. 2 (30 fs, 40 fs), and then the changing phonon spectrum, coupled with electron-phonon interaction, strongly drives the oscillation of charge distribution, as shown in Fig. 6(d,e).
Comment: The theoretical model has been used in many instances in the literature, however precise citations should be given referring to authors that effectively contributed to the model: Su-Schrieffer-Heeger Hamiltonian: cite the original work or a review of the authors; also regarding Brazovskii-Kirova term, Hubbard electron interaction, …. please refer to the appropriate literature. (by the way it is Schrieffer, not Schreiffer).
Response: We appreciate thisyour reminder and have added the related literature [34-37] in the revised version of the manuscript.
Comment: Minor corrections, commenting Figure 2 “We notice the crossing point of levels B and C in is 102 fs…” should be “ We notice the population crossing point of levels B and C is 102 fs…”
Please, illustrate the role of levels A and D in obtaining population inversion between levels B and C
Page 4 line 111: “After conjugated polymers undergo pumping by an external laser with intensity of 70 μJ/cm2..” should be “After the studied conjugated polymers undergo pumping by an external laser with intensity of 70 μJ/cm2….”
Better specify the meaning of: “component of vibrational mode” in Figures 3 and 5.
Response: We are grateful to the reviewer for pointing out these flaws, and we have corrected the manuscript accordingly. The role of levels A and D in obtaining population inversion between levels B and C is added in the corresponding paragraph.
After Figure 1 add:
Especially, the energies of Levels A and B, or Levels C and D, are quite close, indicating very strong non-radiative behavior. As a result, the pumped electrons in Level A can easily jump down to Level B, and the electrons can also easily slip to Level D, which makes this four-level system amenable to population inversion.
The “component of vibrational mode” is just the distribution of the phonon spectrum. This can be obtained by projecting the time-dependent lattice configuration onto the eigenvectors (vibrational mode/phonon modes). Comments on this have been added in the paragraph following Fig. 3.
A Word file is provided for the revised manuscript, with yellow highlights indicating the changes/additions. We hope that this revised version is now acceptable for publication in Molecules.
Sincerely,
The authors
Round 2
Reviewer 2 Report
The paper has been improved and the results are now more clear .
I am happy that the difference between ASE and spontaneous emission has been explained. The paper can now be published.
Reviewer 3 Report
The manuscript “Charge Accumulation of Amplified Spontaneous Emission in a Conjugated Polymer Chain and Its Dynamical Phonon Spectra” has been modified as requested with better explanations and definitions of the performed analysis. Comparison with previously conducted studies and significance/novelty with respect to the previous work is now illustrated. Due references have been added. The work is acceptable in this form.